# Identification of the DPP-IV Inhibitory Peptides from Donkey Blood and Regulatory Effect on the Gut Microbiota of Type 2 Diabetic Mice

**DOI:** 10.3390/foods11142148

**Published:** 2022-07-20

**Authors:** Chaoyue Ma, Dan Liu, Huifang Hao, Xiaotong Wu

**Affiliations:** School of Life Sciences, Inner Mongolia University, Hohhot 010031, China; mcy32108087@163.com (C.M.); ldan153@163.com (D.L.); 111981355@imu.edu.cn (H.H.)

**Keywords:** hemoglobin, DPP-IV inhibitory activity, diabetes, intestinal flora, isolation and purification, molecular docking

## Abstract

After being treated with protease K, peptides extracted from donkey blood were separated, identified, and characterized. The results showed that Sephadex G-25 medium purified with MW < 3 kDa had the highest dipeptidyl peptidase IV (DPP-IV) inhibition capacity. Three-hundred-and-thirty-four peptides were identified with UPLC–MS/MS. Peptide Ranker and molecular docking analysis were used to screen active peptides, and 16 peptides were finalized out of the 334. The results showed that the lowest binding energy between P7(YPWTQ) and DPP-IV was −9.1, and the second-lowest binding energy between P1(VDPENFRLL) and DPP-IV was −8.7. The active peptides(MW < 3 kDa) could cause a reduction in the fasting blood glucose levels of type 2 diabetic mice, improve glucose tolerance, and facilitate healing of the damaged structure of diabetic murine liver and pancreas. Meanwhile, the peptides were found to ameliorate the diabetic murine intestinal micro-ecological environment to a certain extent.

## 1. Introduction

Diabetes mellitus (DM) is one of the most prevalent endocrine and metabolic diseases. Type 2 diabetes mellitus (T2DM) affects 90% to 95% of all diabetic patients [1]. The current study found that the population developing diabetes is gradually becoming younger, and epidemiologists expect that the number of people with T2DM will increase to 439 million by 2030 [2]. Being obese or overweight increases the risks of developing T2DM. T2DM characteristics include insulin resistance and damage to islet β cells [3]. Patients with T2DM often have complications, such as diabetic kidney disease [4], cardiovascular conditions [5], and diabetic cardiomyopathy [6]. Many chronic diseases have been linked to intestinal microbes, perhaps because bioactive metabolites generated by intestinal microflora play a critical role in the occurrence of chronic disorders [7]. Different metabolites produced by gut microbiota are beneficial or harmful to the host intestinal and metabolic homeostasis [8]. Many studies have indicated a significant association between gut microbiota and T2DM. The findings from these studies indicate that the composition and structure of gut microbiota can affect the occurrence of type II diabetes through decreased glucose tolerance and resistance [9].

The current drugs used for T2DM treatment include insulin secretagogues, biguanides, insulin sensitizers, alpha glucosidase inhibitors, incretin mimetics, amylin antagonists, and sodium-glucose co-transporter-2 (SGLT2) inhibitors [10]. However, treatment with these drugs usually causes undesirable side effects. Insulin secretagogues have been developed to treat T2DM by increasing the secretion of insulin from the pancreas, including sulfonylureas and mitiglinides [11]. Treatment with sulfonylureas for T2DM patients usually causes dizziness, sweating, confusion, and nervousness due to low blood sugar levels [12]. Biguanides are used to treat T2DM by causing reductions in the absorption of glucose from the intestine and hepatic glucose output via a decrease in gluconeogenesis and stimulating glycolysis stimulation; meanwhile, biguanides lead to an increase in insulin signaling by increasing insulin receptor activity [13]. Biguanides usually cause adverse effects of gastrointestinal distress, including diarrhea, cramps, nausea, vomiting, and increased flatulence [14]. Glitazones cause an increase in the sensitivity of cells to insulin and lead to a decrease in systemic fatty acid production and fatty acid uptake. Glitazones belong to the peroxisome proliferator-activated receptor (PPARγ) agonists and usually cause various side effects, such as edema, weight gain, macular edema, and/or heart failure [15]. Considering the pathogenesis of T2DM, finding natural products with few side effects has become a crucial study strategy for treating T2DM [16].

Alpha glucosidase inhibitors, incretin mimetics, amylin antagonists, and SGLT2 inhibitors are newly developed drugs for T2DM treatment. Dipeptidyl peptidase-IV (DPP-IV) inhibitors are the most desirable compounds within the group of antidiabetic drugs because of their effectiveness [17]. DPP-IV works as a high-specificity cleaving enzyme that removes X-Pro or X-Ala dipeptides from the N terminus of polypeptides and proteins. It has a strong preference for Pro > Ala > Ser as the penultimate amino acid residue [18,19]. Glucagon-like peptide (GLP) and glucose-dependent insulinotropic polypeptide (GIP) are incretins that control blood sugar by causing an increase in insulin secretion while lowering glucagon secretion in a glucose-dependent manner [20,21]. GLP-1 is the specific substrate of DPP-IV; therefore, DPP-IV inhibitors increase the activity of GLP-1. Most of the DPP-IV inhibitors are peptide derivatives of α-amino acyl pyrrolidines [22]. DPP-IV inhibitory peptides prevent roughly 80–97% of GLP-1 from degradation, extending its half-life from 2 to 40 h [16]. Results from several studies suggest that the actions of initial DPP-IV inhibitors are mediated by GLP-1. When blood sugar control improves, the effect of GIP gradually appears, which causes a further reduction in blood sugar [23].

Donkey blood contains an abundance of proteins and microelements, as well as a modest numbers of minerals, vitamins, hormones, and other bioactive components. Its dry matter includes more than 90% protein, with hemoglobin accounting for 60% to 65% of that value [24]. After enzymatic hydrolysis of hemoglobin, polypeptides with different physiological activities, such as antioxidant [25], antibacterial [26], and DPP-IV inhibitory activities [27], are produced. However, the use of donkey blood has not been given much attention, and only small amounts are collected for research and production. Most donkey blood is discarded as waste, which not only wastes resources but also pollutes the environment.

The polypeptides produced in donkey blood hemoglobin after treatment with protease K were separated, identified, and characterized [28]. In the present study, DPP-IV inhibitory peptides from donkey blood were isolated using ultrafiltration and Sephadex G-25 medium. These DPP-IV inhibitory peptides were sequenced and characterized using ultra-high-pressure liquid chromatography–mass spectrometry (UPLC–MS/MS) and three-dimensional (3D) structural techniques. A molecular docking approach was used to predict the active peptides’ DPP-IV inhibitory activities and explore the sites of action of the active peptides and DPP-IV. The therapeutic impact of the active peptides on T2DM diabetic mice was explored by assessing the active peptides’ influence on changes in the animals’ livers and pancreases. To research the impact of active peptides on the intestinal microorganisms of mice, high-throughput 16S rRNA sequencing technology was used to assess the structure and variety of the intestinal microflora in experimental mice [29]. A theoretical foundation for the utilization of donkey hemoglobin active peptide as a functional food and for the processing and efficient use of donkey blood raw materials is provided.

## 2. Materials and Methods

### 2.1. Materials

Fresh donkey blood was obtained from the Inner Mongolia Hohhot city donkey food designated slaughterhouse. Proteinase K (40 U/mg) was purchased from Merck (Kenilworth, NJ, USA) and a Fast DNA^®^ Spin Kit for Soil was purchased from MP Biomedicals (Irvine, CA, USA). The DPP-IV inhibitor screening kit was obtained from Abnova Co., Ltd. (Walnut, CA, USA) and Sephadex G-25 from Coolaber Co., Ltd. (Beijing, China). All other reagents were commercial products of analytical grade.

### 2.2. Preparation of Hemoglobin Peptides from Donkey Blood

Fresh donkey blood was centrifuged at 5000× *g* for 5 min. The supernatant was discarded and deionized water was added overnight. The mixture was then centrifuged at 15,000× *g* for 15 min and a crude hemoglobin sample was obtained. The extracted crude hemoglobin samples were analyzed for characteristic spectra. Experimental manipulation was performed, according to laboratory-optimized enzymatic conditions. Before adding Proteinase K, we adjusted the crude hemoglobin solution to pH 8. The proteinase K dose was 400 U/g when the substrate mass concentration was 1.0 g/100 mL [30]. Enzymolysis was performed in a water bath at 65 °C for 189 min (DK-8D, Boxun Co., Shanghai, China). After enzymatic hydrolysis, the enzymes were degraded for 10 min at 95.0 °C. After cooling to room temperature, the samples were centrifuged at 15,000× *g* for 15 min, after which the supernatants were collected (5804R, Eppendorf Co., Ltd., Hamburg, Germany). The use and disposal of donkey blood met the standards of the Ethics Committee of Inner Mongolia University.

### 2.3. DPP-IV Inhibitory Peptide Activity Assay

We selected a fluorescence approach, which involved adding chemicals according to the DPP-IV inhibitor screening kit’s instructions, followed by incubation at 37 °C for 30 min, after which fluorescence intensity was detected at 360 and 460 nm. Using formula (1), the inhibition rate was estimated:(1)$$DPP-IV/\%=\left(1-\frac{a-b}{c-b}\right)*100$$
where a = the fluorescence intensity of the inhibitor wells; b = the fluorescence intensity of the background wells; and c = the fluorescence intensity of the DPP-IV wells.

### 2.4. Preparation of Hemoglobin Peptides from Donkey Blood

#### 2.4.1. Ultrafiltration

Following protease K hydrolysis, samples of donkey hemoglobin were progressively ultrafiltered using centrifugal filters with 6 and 3 kDa MWCO. Recovered fractions (sample I, 3–6 kDa; sample II, MW < 3 kDa) were freeze-dried and stored at −80 °C for later use.

#### 2.4.2. Sephadex G-25 Medium

The active ingredients were purified using a Sephadex G-25 dextran gel column. At a flow rate of 1 mL/min, distilled water was used to elute 2 mL of the samples (40 mg/mL). Fractions were then collected for later use. Component III (UC-III) was purified from component II (UC-II) fragment using Sephadex G-25 medium. Measurement of the DPP-IV inhibitory activities of UC-I–III was performed as described in Section 2.3.

#### 2.4.3. UPLC–MS/MS-Based Peptide Identification

The portion with the strongest 2,2-diphenyl-1-picrylhydrazyl (DPPH) radical scavenging activity was sequenced using ultra-high-performance liquid chromatography–mass spectrometry (UPLC–MS/MS). The Ultimate 3000 was used for chromatographic separation. The peptides were dissolved in 0.1% formic acid mixed with 2% acetonitrile and 98% deionized water and then loaded onto a C18 trap column at a flow rate of 600 nL/min, with a 66 min gradient. The mobile phase consisted of acetonitrile/formic acid/water (2/0.1/98, *v*/*v*/*v*) for buffer A and acetonitrile/formic acid/water (80/0.1/20, *v*/*v*/*v*) for buffer B. Elution was performed according to a linear elution procedure (Table 1). After the mass spectrometry assay was complete, the resulting raw file was retrieved from the target protein database using Byonic.

### 2.5. Molecular Docking

From the Protein Data Bank (PDB) database, we downloaded and pretreated the crystal structure of human DPP-IV/CD26 (PDB ID: 1n1m) using AutoDock Tools 1.5.6. Two-dimensional (2D) structure models of the identified peptides were predicted using ChemDraw 20.0 and by applying the minimum MM2 force field energy. The receptor (DPP-IV) was set as rigid and the ligands (the identified peptides) were set as flexible. Based on –CDOCKER interaction energy (–CIE) score, interaction site, and interaction of four types with DPP-IV, the molecular docking results were evaluated.

### 2.6. Effects of DPP-IV Inhibitory Peptides on Blood Sugar and Intestinal Microflora of Mice

#### 2.6.1. Modeling of Mice

Out of 60 ICR mice, 10 were randomly selected as the normal control group (CK) and fed with normal chow. The remaining 50 mice were injected intraperitoneally with streptozotocin (STZ) for three days after being fed a high-fat diet for four weeks. Five days after withdrawal of the drug, blood was taken from the fasting tail vein to measure blood glucose levels. A blood glucose level greater than 11 mmol/L [31] indicated that the T2DM model in mice was successful. After the diabetic mice were randomly divided into five groups, they received 0.2 mL of their respective drug doses (Table 2), and the treatment was continued for 28 days. The animal ethics of the experiments described above met the standards of the Ethics Committee of Inner Mongolia University.

#### 2.6.2. Blood Glucose Testing

After the last treatment, blood for glucose measurements was collected from the tails of the mice, which had been starved for 8 h. Mice received a 2 g/kg glucose solution intra-gastrically, and blood was obtained from the tails of the mice in each group at 30, 60, and 120 min after glucose administration to determine blood glucose levels. The area under the curve (AUC) of each group was computed using the method described below based on the blood glucose concentration detection data for each time point:$$\mathrm{AUC}=\left(\mathrm{FBG}+\mathrm{BG}30\;\min\right)1/4+\left(\mathrm{BG}30\;\min+\mathrm{BG}60\;\min\right)1/4+\left(\mathrm{BG}60\;\min+\mathrm{BG}120\;\min\right)1/2$$
where FBG = fasting blood glucose; and BG = blood glucose.

#### 2.6.3. Histological Examination

The mice were sacrificed, and liver and pancreatic histopathology was performed. Using phosphate-buffered saline (PBS), the livers and pancreases were washed, and the colonic mucosae were collected on a microscope slide. The liver and pancreatic tissues were washed and then fixed in 4% formalin, embedded in paraffin, and sectioned [32]. Hematoxylin and eosin (H&E) was used to stain the specimens after sectioning.

#### 2.6.4. 16s rRNA Gene Sequencing

Three mice in each group were randomly selected for the isolation of cecal contents under aseptic conditions and microbial flora were analyzed using 16s rRNA gene sequencing. Total bacterial DNA was extracted from the intestinal contents of the mice using the FastDNA^®^ Spin Kit II (MP Biomedicals, Irvine, CA, USA). Using the extracted DNA, 16S rRNA genes were amplified and sequenced using next-generation sequencing with the Illumina platform (Illumina, San Diego, CA, USA) [33,34]. To generate amplicon sequence variant tables, a mix of analytic programs, including PANDAseq v2.8 and QIIME v2.0 for 16S rRNA gene pre-processing [35,36], was used. The GreenGenes V13.8 database was used to perform the classification.

#### 2.6.5. GC–MS

After the last dose, six mice were randomly selected from the CK and NC groups for the collection of feces under sterile conditions, while two mice were randomly selected from each of the HEH, HEM, and HEL groups to be mixed with the HE group. After the mouse fecal samples were processed, samples were separated using a DB-5MS capillary column (40 m × 0.25 mm × 0.25 μm) and then subject to mass spectrometric detection.

### 2.7. Statistical Analysis

All trials were carried out in triplicate. Microsoft Excel 2020(Microsoft Co., Redmond, WA, USA) was used to create tables, and Origin v8.0 (OriginLab, Northampton, MA, USA) and Microsoft PowerPoint 2016(Microsoft Co., Redmond, WA, USA) were used to create figures. SAS software v8 was used to analyze variance and regression (SAS Institute Inc., Carry, NC, USA). Duncan’s multiple range test was used to determine differences between mean values. Significant differences were confirmed when *p* < 0.05.

## 3. Results

### 3.1. Preparation of Hemoglobin Hydrolysates from Donkey Blood

We identified the extracted donkey hemoglobin samples based on the characteristic spectral scan. The results showed that the samples exhibited absorption peaks at 275, 346, 417, 542, and 577 nm (Figure 1), similar to the results reported for standard hemoglobin [19]. These results indicated that the donkey hemoglobin had been successfully extracted. Hemoglobin hydrolysates were prepared by incubation with proteinase K. The DPP-IV inhibitory effects of the enzymatic products were examined using a DPP-IV inhibitory peptide activity assay. The inhibition of DPP-IV by 2 mg/mL enzymatic product was 53.44% under the enzymatic conditions measured in this study. This finding indicates that donkey blood hemoglobin hydrolysate had an inhibitory effect on the activity of DPP-IV.

### 3.2. The Hemoglobin Hydrolysates from Donkey Blood Inhibit the Activity of DPP-IV

The hemoglobin hydrolysates of donkey blood exhibited certain DPP-IV inhibitory effects; therefore, the hemoglobin hydrolysates were further purified to identify the DPP-IV inhibitory peptides by ultrafiltration using a Sephadex G-25 dextran gel column. The DPP-IV inhibitory activities of ultrafiltration component I (UC-I, molecular weight 3–6 kDa) and ultrafiltration component II (UC-II, molecular weight < 3 kDa) from the hemoglobin hydrolysates of donkey blood were determined. Component II was purified with Sephadex G-25 medium, and component III (UC-III) was collected and used to measure its DPP-IV inhibitory activity. The results showed the highest DPP-IV inhibitory activity for UC-III (IC_50_ = 1.95 ± 0.39 mM; *p* < 0.05), followed by UC-II (3.58 ± 0.58 mM; *p* < 0.05). UC-I (IC_50_ = 4.24 ± 0.46 mM; *p* < 0.05) exhibited the lowest DPP-IV inhibitory activity among all fractions (Figure 2).

### 3.3. Identification of DPP-IV Inhibitory Peptides by UPLC–MS/MS

We selected fraction III with the highest DPP-IV inhibitory activity for identification by LC–MS/MS, and polypeptides were examined using primary and secondary mass spectrometry-based molecular ion peaks. Three-hundred-and-thirty-seven peptide sequences were obtained with high confidence in relation to DPP-IV inhibitory activity and amino acids, after which we further selected 16 peptides by predicting the probability of active peptides with biological activity using the Peptide Ranker online software. The secondary spectra of the 16 peptides are shown in Appendix A, and the amino acid sequences are shown in Table 3.

### 3.4. Molecular Docking Analysis of Identified Peptides with DPP-IV

We selected 16 peptides with high biological activity for molecular docking analysis based on Peptide Ranker score predictions. The lower binding energy of the peptides to DPP-IV indicated more stable binding. The bioactivity prediction and molecular docking results (Table 4) showed that P1(VDPENFRLL) had a peptide sequence score of 0.74 and a tight binding energy of −8.70; DPP-IV P7(YPWTQ) had the lowest binding energy and the tightest structure, with a binding energy of −9.1 and a peptide sequence score of 0.66 with DPP-IV. Molecular docking is mainly achieved through intermolecular interactions that force molecular recognition by simulating the interactions of a large-molecule protein receptor with a small-molecule peptide ligand to analyze the interaction site and the mode of interaction between them. The molecular structural formulae for the active peptides P1 and P7 were drawn in ChemDraw 20.0, and 3D structures were obtained after energy minimization, as shown in Table 5. The optimized peptides were docked with DPP-IV. We used AutoDock to complete the docking (Figure 3), and it was found that the P1 and P7 ligands were in close contact in the 3.5 Å range of DPP-IV amino acid residues (Table 5). Six amino acid residue sites closer to DPP-IV were identified for peptide P1, namely, His-748 (2.4 Å), Tyr-752 (2.0 Å), Tyr-48 (2.1 Å), Lys-554 (2.3 Å), Gln-553 (2.7 Å), Tyr-547 (2.1 Å /2.7 Å), and nine amino acid residue sites were identified as being closer to DPP-IV for peptide P7, namely, Tyr-195 (2.4 Å), Tyr-211 (2.4 Å /2.7 Å), Trp-124 (2.2 Å), Asp-709 (2.4 Å), Ala-707 (2.5 Å), Tyr-238 (2.4 Å), Arg-253 (2.1 Å), Lys-122 (1.9 Å), and Gln-123 (2.3 Å). It can be seen that the peptide P7 has many binding sites and short distances, which was consistent with the binding energy. Wang et al. reported that the N-terminal second residues of oligopeptides with DPP-IV inhibitory activity are generally Pro, Trp, Ala, Val, Lys, and Asp, with Pro being the most preferred [37]. This finding may be one of the reasons for the low binding energy of P7 to DPP-IV. In addition, peptide P1 exhibits the second-lowest binding energy to DPP-IV; one possible reason could be that the second amino acid residue at its N-terminal end is Asp. A reference sequence for future studies of DPP-IV inhibited synthetic peptides is needed.

### 3.5. DPP-IV Inhibitory Peptides Cause Reductions in the Fasting Glucose Levels of Type 2 Diabetic Mice

The DPP-IV inhibitory activity of purified active peptides from donkey blood was confirmed by DPP-IV inhibitory peptide activity assay and molecular docking analysis. Next, we examined the hypoglycemic effects of purified active peptides from donkey blood on the blood glucose levels of type 2 diabetic mice.

Based on an analysis of the changes in fasting blood glucose (FBG) of the mice in each group (Table 6), we found that the blood glucose levels of mice in the other five groups (NC, SP, HEL, HEM, and HEH) were substantially different from those of the CK group at the start of the trial (0 d). At the end of the experiment (28 d), the blood glucose levels of mice in the other four groups, except for the CK group, decreased to a certain extent and showed extremely significant differences when compared with the NC group (*p* < 0.01). Through analysis of the changes in glucose tolerance of mice in each group (Table 7), we demonstrated that the blood glucose of mice in each group peaked rapidly 0.5 h after glucose injections (2 g/kg). After 2 h, the blood glucose levels of all the groups decreased, except for the levels in the NC group, which remained high. The active peptide intake group had a considerably lower area under the curve (AUC) value than the NC group (*p* < 0.05), showing that the active peptide may lead to an effective reduction in postprandial blood glucose increase in diabetic mice and produce a significant effect in terms of increasing glucose tolerance in mice.

### 3.6. Histopathology

It can be seen from the H&E staining of mouse liver pathological sections (Figure 4A) that the animals in the CK group had decent tissue structure. The general structure of the tissues in the NC group was somewhat aberrant, the structure of the liver cells was loose, a small number of liver cells in the tissues were slightly edematous, and hepatic sinusoids showed signs of congestion. The liver tissue in HEL had a moderately aberrant structure, overall. The liver cells had a complete structure, the central vein was visible, and the hepatic sinusoids were dilated. The HEM and HEH groups’ liver tissue structures and shapes were dramatically better than those of the NC group. This finding indicates that the active peptide can facilitate the repair of liver damage in diabetic mice to a certain extent, but this process is affected by the active peptide dose.

It can be seen from the H&E staining of mouse pancreas pathological sections (Figure 4B) that the CK group showed no visible edema, fatty degeneration, necrosis, or other degeneration of the acinus, and no clear inflammatory cell infiltration in the tissue was detected. Compared with the NC group, the overall structure of the pancreatic tissue in the HEL group was slightly abnormal, and no obvious inflammatory cell infiltration in the tissue was found. The improvement in pancreatic tissue structure was more noticeable in the SP, HEL, HEM, and HEH groups than in the NC group. The islet cells were arranged regularly and were abundant, the structure of the acinar epithelial cells was full, no obvious degeneration, such as loose edema and necrosis, was noted, and no obvious inflammatory cell infiltration had occurred in the tissue in the HEH group’s pancreases. With an increase in dosage, the damage recovery effect in the diabetic mice was enhanced.

### 3.7. Effects of DPP-IV Inhibitory Peptides on the Intestinal Microflora of Mice

Many studies have demonstrated that the development of diabetes is related to changes in the compositional profile of gut microbiota [38]. Our data indicated that the isolated and purified DPP-IV inhibitory peptides from donkey blood reduced the fasting glucose of type 2 diabetic mice. The hypoglycemic effects of DPP-IV inhibitory peptides might be due to the regulation of gut microbiota. In order to explore the possible regulatory effects of DPP-IV inhibitory peptides on gut microbes, the microbial flora of type 2 diabetic mice in different treatment groups were analyzed using 16s rRNA gene sequencing.

#### 3.7.1. Sample Diversity Analysis

The Illumina Miseq platform was used to assess data quality, and the sequencing findings revealed a total of 1,142,804 valid sequences, with 476,013,057 valid bases and an average sequence length of 416. The sample dilution curves (Figure 5A) and Shannon exponential curves (Figure 5B) were both relatively flat, indicating that the number of sequencing data was reasonable and that the sample’s sequencing depth was sufficient. The rank abundance curve (Figure 5C) [39] decreased smoothly, indicating that the samples were diverse. The other five groups’ Shannon, Ace, and Chao1 indices (Figure 6A,C,D) were greater than those of the NC group, while the opposite was the case for the Simpson indices (Figure 6B), showing that polypeptide consumption improved the variety and quantity of microorganisms in the guts of mice.

#### 3.7.2. Intestinal Flora

The results regarding intestinal flora are shown in Figure 7. At the family level, the intestinal flora of diabetic mice significantly increased (NC: 43.21%; CK: 25.08%; *p* < 0.05), and representation of the *Lactobacillus* family was significantly reduced (NC: 10.38%; CK: 19.95%; *p* < 0.05). However, the abundance of intestinal flora in the *Lactobacillus* family of diabetic mice after the intervention with the active peptides in different dose groups significantly increased (HEL: 16.55%; HEH: 18.85%; NC: 10.38%; *p* < 0.01), the abundance of Lachnospiraceae decreased significantly (HEL: 37.71%; HEM: 20.65%; HEH: 33.72%; NC: 45.81%; *p* < 0.05). At the family level, the active peptide therapy group’s intestinal flora tended to be restored to the normal intestinal microbial makeup of mice. It has been reported in the literature that *Lactobacillus*, an essential probiotic, is beneficial for preserving intestinal barrier function, increasing intestinal immunity, and preventing and treating metabolic illnesses, including obesity and diabetes [40,41]. In this study, the active peptide could lead to a significant increase in the abundance of *Lactobacillus* in the intestinal microbial composition of diabetic mice. Cui et al. reported that berberine could exert a hypoglycemic effect by reducing the abundance of Lachnospiraceae [42]. The intestinal microbial composition of diabetic mice changed similarly after treatment with the active peptide in our investigation. These findings suggested that the polypeptide’s hypoglycemic impact was linked to its modulation of the intestinal microorganisms in diabetic mice.

#### 3.7.3. Analysis of Differences between Sample Groups

The colony makeup of the CK and NC groups differed, demonstrating that diabetes could lead to alterations in the intestinal flora of mice according to hierarchical clustering analysis (Figure 8A). However, compared to the CK group, the change in colony composition following polypeptide therapy was minimal, demonstrating that the active peptides may facilitate repair and reconstruction of intestinal flora composition and structure. In the principal coordinates analysis (PCoA; Figure 8B), samples from the NC group were generally grouped in one area based on aggregation, whereas the HEL, HEM, and HEH groups tended to be close to the CK group. This finding indicates that the intestinal microbial composition of diabetic mice improved dramatically after receiving the active peptide and resembled the intestinal microflora of normal mice.

#### 3.7.4. The Effects of Active Peptides on Intestinal Microbial Metabolites

We found intestinal metabolites, including mainly vitamins and coenzyme factors, peptides, nucleic acids, hormones and transmitters, steroids, organic acids, lipids, and carbohydrates (Figure 9A), in the feces of the different groups of mice. The partial least squares discriminant analysis (PLS-DA), shown in Figure 9B, of microbial metabolites indicated that some differences between the metabolites in the CK and NC groups existed. The metabolites in the HE group were partially segregated from those in the NC group, suggesting that the active peptides could cause alterations in microbial metabolite levels in diabetic mice. Furthermore, the partial metabolites in the NC group were similar to those in the CK group, demonstrating that the active peptide might lead to partial restoration of microbial metabolism in mice [43].

The seven metabolites in variable importance in projection (VIP) value diagrams of the PLS-DA model (Figure 9C) revealed significant differences, namely, 2-phenylacetamide, 2,8-quinoline diol, carbazole, lactic acid, L-threonine, pyrogallol, and benzene-1,2,4-triol. The levels of 2-phenylactamide, 2,8-quinolinediol, and carbazole were reduced in the HE group, whereas the amounts of lactic acid, L-threonine, pyrogallol, and benzene-1,2,4-triol dramatically increased. In previous studies, *Lactobacillus* has been shown to have an essential function in controlling the balance of intestinal microbes, improving immunity, facilitating digestion, and lowering cholesterol [44]. In this study, we found that the alterations in mouse metabolites matched those in gut flora. Lactic acid contents increased as *Lactobacillus* populations increased [45]. Lactic acid infiltrated pathogenic bacteria and accumulated in cells as a result of the increase in lactic acid, thus lowering the pH of the intestinal environment and suppressing pathogenic bacteria. These metabolites also interacted with phospholipid molecules, lipopolysaccharide, and other components to compromise membrane stability.

## 4. Discussion

Diabetes has become a global public health problem and a threat to human health [46,47]. The drugs currently on the market to treat diabetes have the capability of lowering blood glucose, eliminating clinical symptoms, and delaying the onset of complications for a short period of time. However, long-term use of these drugs can also lead to drug resistance. Researchers are looking for options with fewer side effects that can control blood sugar levels and reduce diabetes complications [48]. It has become popular to study DPP-IV inhibitory peptides from food protein sources. In this study, DPP-IV inhibitory peptides extracted from donkey blood reduced fasting blood glucose levels and improved glucose tolerance in mice with type 2 diabetes. The abnormal morphology of the liver and pancreas in diabetic mice was restored to some extent.

Since 2006, DPP-IV inhibitors (DPP-IVis) have been available for the treatment of T2DM. These inhibitors have certain advantages as oral hypoglycemic agents, such as weight neutrality, less risk of hypoglycemia, and an insulin-independent mechanism of action as an oral hypoglycemic agent, when compared with other drugs, such as sulfonylureas [49]. Although DPP-IVis have good therapeutic efficacy and are safe and effective in most patients with T2DM, uncertainties remain regarding the effects of DPP-IV inhibitors on cardiovascular disease, pancreatitis, and other issues [50]. Compared with synthetic drugs, the search for functional foods with natural and minimal side effects has become a hot research topic. Currently, effective DPP-IV inhibitory peptides are widely derived from food proteins, such as milk. Wang et al. [51] found that a Micropterus salmoides enzymatic digest could inhibit DPP-IV activity. The peptide VAGTWY, which inhibits DPP-IV, was isolated from an enzymatic digest of β-lactoglobulin from Haute Cheese and finally synthesized. Gelatin prepared from animal skin collagen, which is rich in Pro, also has DPP-IV inhibitory activity in its hydrolysate. In addition, plant products derived from *Avena sativa* L., *Fagopyrum esculentum*, and *Hordeum vulgare trifurcatum* (L.) Trofim are also good protein sources for the preparation of DPP-IV inhibitory peptides [52].

We proposed using the DPP-IV inhibitory peptides from donkey blood in this study after considering that donkey blood is rich in proteins and trace elements; in addition, small amounts of minerals, vitamins, hormones, and other bioactive components are also found in donkey blood. Very few protocols for the utilization of donkey blood are available. We prepared DPP-IV inhibitory peptides using proteinase K enzymatic digestion of donkey blood. This peptide can lower blood glucose, improve glucose tolerance, lead to the restoration of abnormal liver and pancreatic morphology to some extent and can also restore intestinal microbial diversity in diabetic mice (Figure 10). DPP-IV has natural and safe characteristics. The DPP-IV food protein-derived peptide inhibitors identified so far have not shown the same efficacy as synthetic drug inhibitors. However, these inhibitors may be best suited for combination therapy in patients with prediabetes or mild diabetes [53]. Research on potential functional components with DPP-IV inhibitory properties is expected to be applied in the treatment or control of the onset and progression of T2DM. Therefore, the preparation of DPP-IV inhibitory peptides from donkey blood for the treatment of T2DM shows promise regarding future applications.

We attempted to elaborate the mechanism of the DPP-IV inhibitory peptides prepared in this study for the treatment of T2MD with respect to gut microbes. It was reported before this study that dysregulation of gut microbial populations may reshape gut barrier function and host metabolic and signaling pathways, which are directly or indirectly associated with insulin resistance in T2DM. Moreover, changes in the numbers and diversity of intestinal flora are important for the progression of many metabolic disorders [54,55]. In our experiments, we successfully established a type 2 diabetic mouse model using high-fat chow and found that T2MD leads to a significant reduction in intestinal microbial diversity in mice [56]. Previous studies have reported that probiotics exhibit beneficial effects on insulin antagonism in animal models of diabetes. However, probiotics also have an important role in assisting other intestinal microbes to maintain the integrity/homeostasis and metabolism of the intestinal epithelial lining [57,58]. For example, treatment with *Lactobacillus* fermentum MTCC 5689 has been shown to improve insulin resistance and prevent the development of diabetes in HFD-induced diabetic mice [59]. In addition, administration of *L. paracasei* TD062 produced an improvement in glucose homeostasis and an enhancement in insulin signaling pathways, preventing the development of T2DM [60]. Therefore, when we found a significant increase in the abundance of *Lactobacillus* spp. in the gut microbes of diabetic mice after peptide treatment, we felt that we had obtained evidence of the benefits of the use probiotics in the treatment of diabetic patients. *Lachnospiraceae_NK4A136_group*, representing a butyrate-producing bacterium, has been found to maintain the integrity of the intestinal barrier in mice and was negatively correlated with intestinal permeability [61,62]. In addition, it was previously reported that flavopiridol, which has hypoglycemic properties, reduced the abundance of Trichophytonaceae in the intestinal microbes of diabetic mice. The decline in glucose levels and restoration of intestinal microbial diversity in diabetic mice after peptide treatment may be related to the reduced abundance of *Lachnospiraceae_NK4A136_group*.

In addition to the brief description of the metabolite lactate in Section 3.7.4, we discussed the significance of metabolites other than lactate in this study. In the study on intestinal metabolites, it was reported that 6-formylindolo (3,2-b) carbazole-induced activation of aryl hydrocarbon receptors prevented intestinal barrier dysfunction by regulating claudin-2 expression [63]. It has been suggested that carbazole plays a role in restoring intestinal barrier function. It has also been reported that some polyphenolic metabolites produced in the colon are capable of counteracting in vitro protein glycosylation, a key feature of diabetic complications [64]. This finding suggests that o-threitol may play an important role in hypoglycemia in vivo. In addition, although no studies concerning the relationship between L-threnine, benzene-1,2,4-triol, and the intestine are available, it can be inferred from their properties that they have an important role in maintaining the intestinal barrier. Regarding the selection of animals for this study, only male mice were used, so that future studies using female animals are needed. In addition, although 16 peptides with high biological activity were identified by computer simulation in this study, they were not validated for in vitro DPP-IV inhibitory activity and related cellular assays. These assays could provide peptide sequences for the 16 DPP-IV inhibitory peptides screened in this study by solid-phase synthesis in the future. Further studies could also be conducted on L-threonine and benzene-1,2,4-triol in relation to the intestinal tract, which have not currently been studied.

## 5. Conclusions

In this work, we identified and characterized DPP-IV inhibitory peptides generated by protease K hydrolysis of donkey blood samples. DPP-IV inhibitory peptides were separated, identified, and characterized using ultrafiltration, RP-HPLC, DPP-IV inhibitory peptide activity, UPLC–MS/MS, three-dimensional (3D) structure in silico prediction and molecular docking analysis. Among the identified peptides, P7 showed the lowest DPP-IV binding energy of −9.1. The second-lowest binding energy between P1 and DPP-IV was −8.7. The mixed peptide (MW < 3 kDa) isolated in this study could lead to a reduction in fasting blood glucose levels in type 2 diabetic mice and improvement in glucose tolerance. The intake of the active peptides can lead to the restoration of the abnormal morphology of the liver and pancreas in diabetic mice to a certain extent. Active peptides restore intestinal microbial diversity in diabetic mice. This study provides a reference for the follow-up development of high value-added products of donkey blood.

## Figures and Tables

**Figure 1 foods-11-02148-f001:**
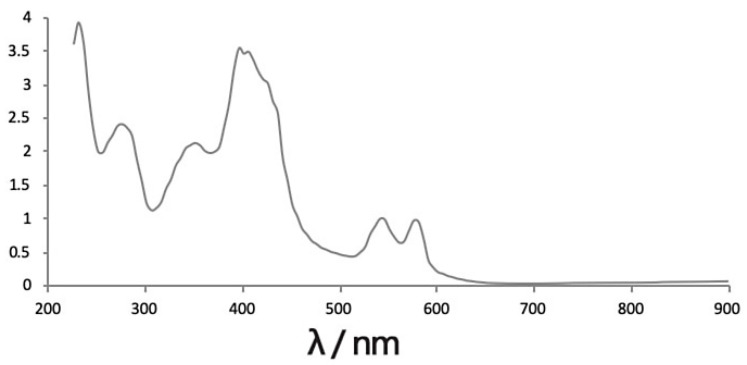
Scanning diagram of the total wavelength for donkey hemoglobin.

**Figure 2 foods-11-02148-f002:**
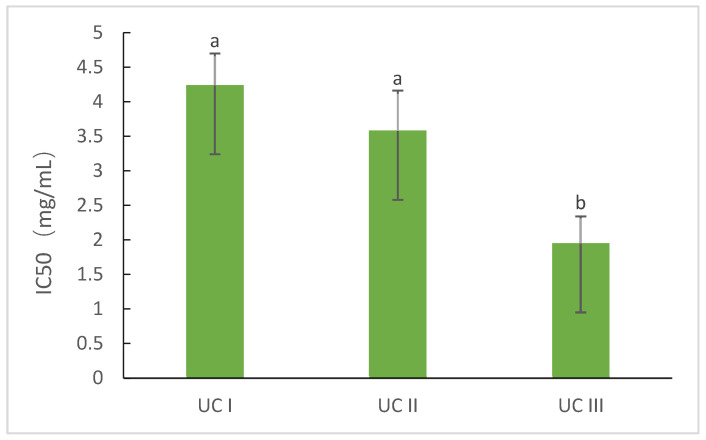
Dipeptidyl peptidase IV (DPP-IV) inhibitory activity of each fraction. Different letters indicate significant differences (*p* < 0.05).

**Figure 3 foods-11-02148-f003:**
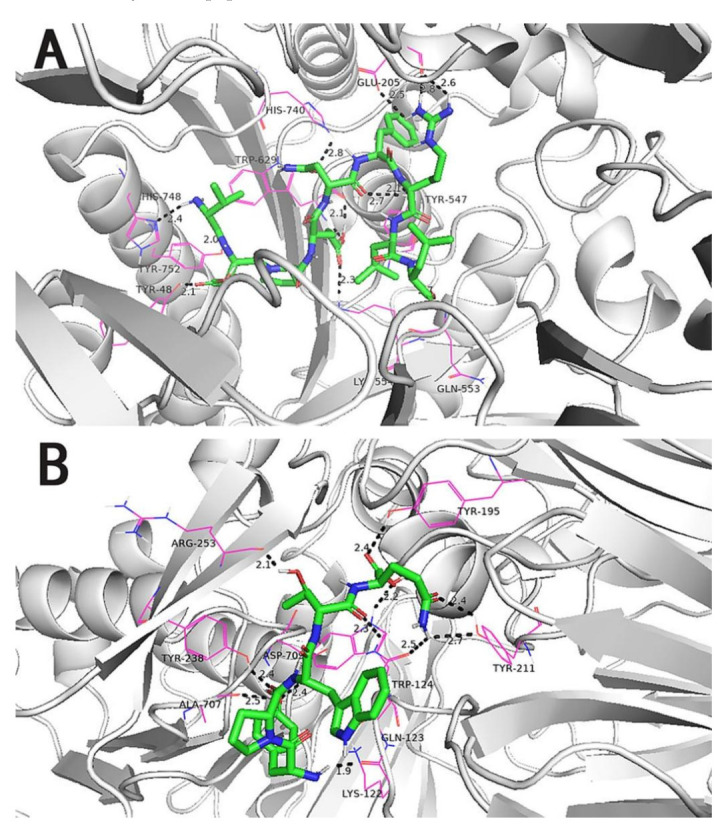
(**A**) Molecular docking diagram of VDPENFRLL and DPP-IV. (**B**) Molecular docking diagram of YPWTQ and DPP-IV.

**Figure 4 foods-11-02148-f004:**
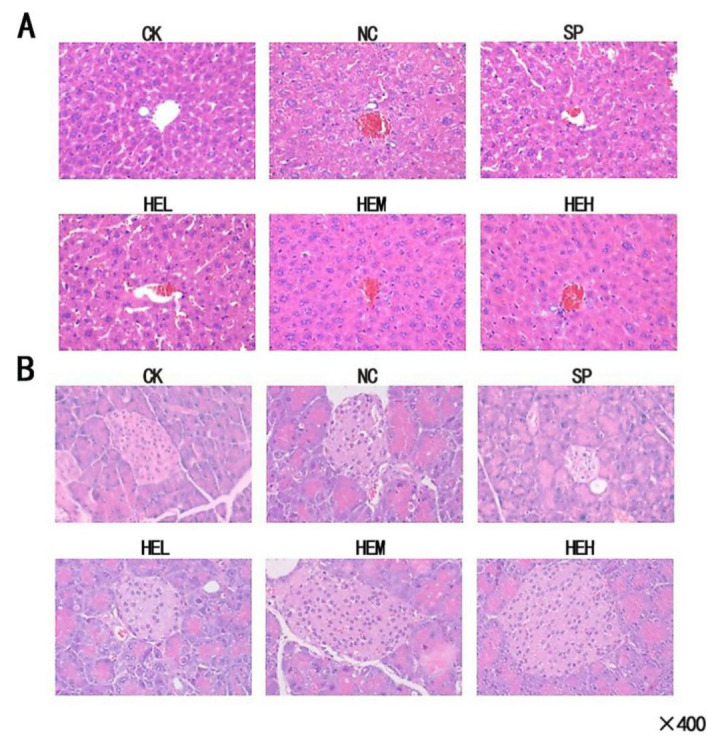
(**A**) Liver histology (hematoxylin and eosin (H&E) stain) in mice (×400). (**B**) Pancreas histology (H&E stain) in mice (×400). CK: Blank control group; NC: Model control group; PC: Sitagliptin group; HEL: Active peptide low-dose group; HEM: Active peptide medium-dose group; HEH: Active peptide high-dose group.

**Figure 5 foods-11-02148-f005:**
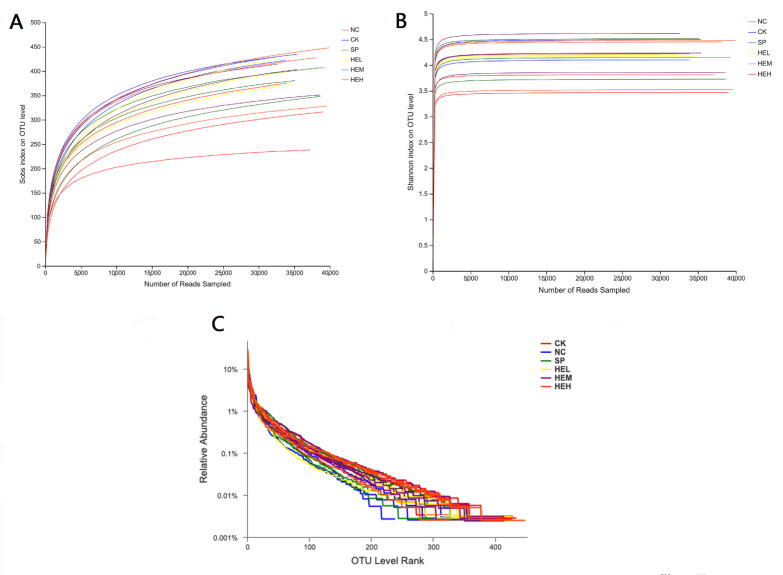
Analysis of samples and diversity indices of mice in different treatment groups. (**A**) Sobs curve. (**B**) Shannon index curve. (**C**) Rank abundance.

**Figure 6 foods-11-02148-f006:**
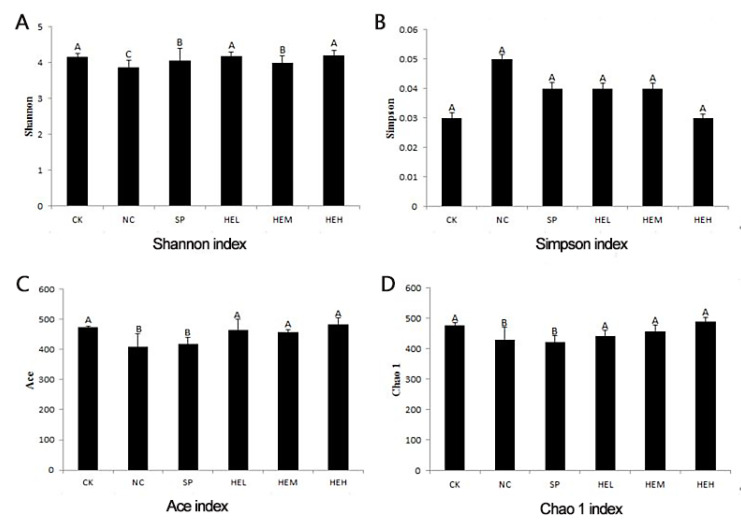
(**A**) Shannon index. (**B**) Simpson index. (**C**) Ace index. (**D**) Chao 1 index. Different letters indicate significant differences (*p* < 0.05).

**Figure 7 foods-11-02148-f007:**
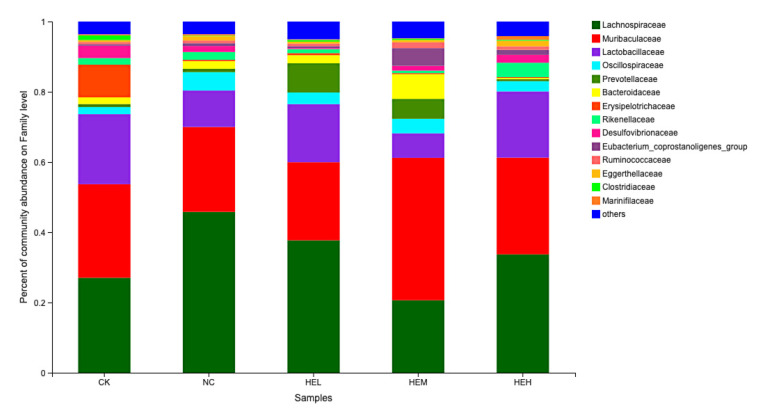
The effect of different treatments on the composition of mice gut microbes. Histogram of family-level communities.

**Figure 8 foods-11-02148-f008:**
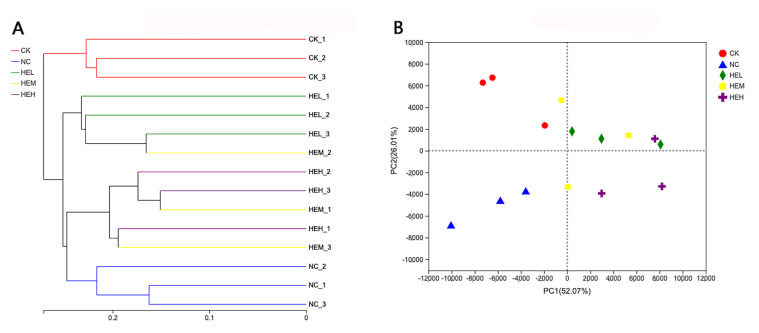
(**A**) Hierarchical clustering. (**B**) Principal Coordinates Analysis (PCoA) of different samples.

**Figure 9 foods-11-02148-f009:**
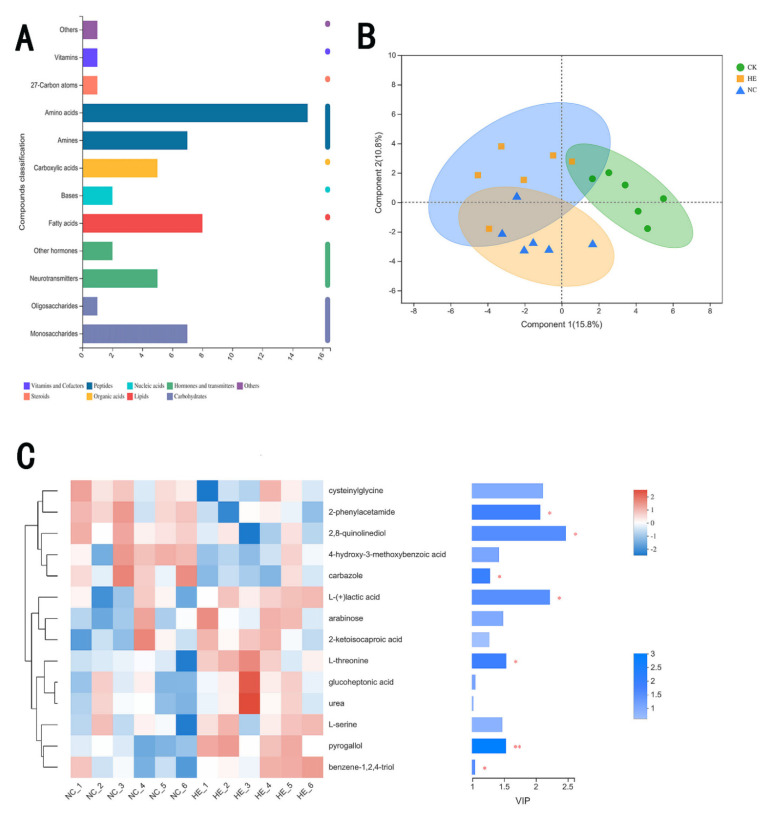
(**A**) Kyoto Encyclopedia of Genes and Genomes (KEGG) compound classification statistics. (**B**) PLS−DA analysis of microbial metabolites in the feces of the mouse groups. (**C**) Variable importance in projection (VIP) analysis of fecal microbial metabolites in the NC and HE groups. CK: Blank control group; NC: Model control group; HE: Two mice were randomly selected from each of the HEH, HEM, and HEL groups to be mixed. Compared with the NC group, * *p* < 0.05, ** *p* < 0.01.

**Figure 10 foods-11-02148-f010:**
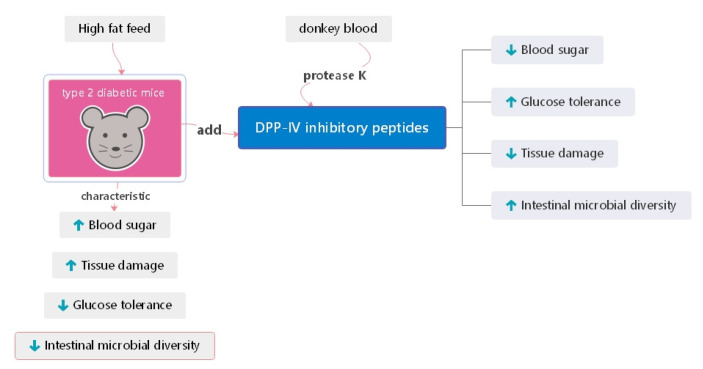
Experimental general mode diagram.

**Table 1 foods-11-02148-t001:** Liquid chromatography gradient elution program.

Time (min)	Buffer B	Time (min)
0	4%	0
2	8%	2
45	28%	45
55	40%	55
56	95%	56

**Table 2 foods-11-02148-t002:** Grouping of mice.

Group	Number	Administration Dosage
Blank control group (CK)	10	Saline
Model control group (NC)	10	Saline
Sitagliptin group (PC)	10	15 mg/kg Sitagliptin
Active peptide low-dose group (HEL)	10	400 mg/kg polypeptide
Active peptide medium-dose group (HEM)	10	800 mg/kg polypeptide
Active peptide high-dose group (HEH)	10	1200 mg/kg polypeptide

**Table 3 foods-11-02148-t003:** Peptide Ranker score and binding energy data from the molecular docking analysis.

No.	Peptide	*m*/*z*	Z	Mass
P1	VDPENFRLL	551.80	2.00	1102.59
P2	HLPNDF	371.68	2.00	742.35
P3	LPGAL	470.29	1.00	470.29
P4	HVDPENFRLL	620.33	2.00	1239.65
P5	VDPENFRL	495.26	2.00	989.51
P6	VDPVNFKLL	522.81	2.00	1044.61
P7	YPWTQ	694.32	1.00	694.32
P8	PVNF	476.25	1.00	476.25
P9	HVDPENFRL	376.19	3.00	1126.56
P10	VDPVNFKL	466.27	2.00	931.53
P11	DLPGAL	585.33	1.00	585.33
P12	FPHFDLSHGS	572.26	2.00	1143.52
P13	HLPNDFTP	470.73	2.00	940.45
P14	DFTPA	550.25	1.00	550.25
P15	PHFDLS	358.17	2.00	715.34
P16	LPNDFTPA	437.72	2.00	874.43

A: Ala; V: Val; L: Leu; I: Ile; F: Phe; W: Trp; M: Met; P: Pro; G: Gly; S: Ser; T: Thr; C: Cys; Y: Tyr; N: Asn; Q: Gln; H: His; K: Lys; R: Arg; D: Asp; E: Glu.

**Table 4 foods-11-02148-t004:** Peptide Ranker score and binding energy data for docking.

No.	Peptide	Peptide Sequence Score	DPP-IV Binding Energy
P1	VDPENFRLL	0.74	−8.70
P2	HLPNDF	0.72	−7.50
P3	LPGAL	0.71	−7.60
P4	HVDPENFRLL	0.70	−8.00
P5	VDPENFRL	0.70	−7.80
P6	VDPVNFKLL	0.69	−8.20
P7	YPWTQ	0.66	−9.10
P8	PVNF	0.66	−7.80
P9	HVDPENFRL	0.65	−8.10
P10	VDPVNFKL	0.62	−8.60
P11	DLPGAL	0.62	−8.20
P12	FPHFDLSHGS	0.61	−8.10
P13	HLPNDFTP	0.58	−8.40
P14	DFTPA	0.58	−7.70
P15	PHFDLS	0.54	−8.50
P16	LPNDFTPA	0.53	−7.30

A: Ala; V: Val; L: Leu; I: Ile; F: Phe; W: Trp; M: Met; P: Pro; G: Gly; S: Ser; T: Thr; C: Cys; Y: Tyr; N: Asn; Q: Gln; H: His; K: Lys; R: Arg; D: Asp; E: Glu.

**Table 5 foods-11-02148-t005:** Molecular formulae and three-dimensional (3D) polypeptide structures.

NO.	Sequence	Interactions with DPP-IV	3D Polypeptide Structures
P1	VDPENFRLL	His-748, Tyr-752, Tyr-48, Lys-554, Gln-553, Tyr-547	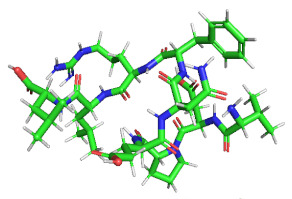
P7	YPWTQ	Tyr-195, Tyr-211, Trp-124, Asp-709, Ala-707, Tyr-238, Arg-253, Lys-122, Gln-123	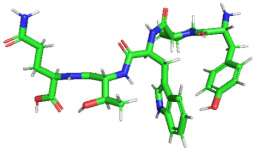

**Table 6 foods-11-02148-t006:** Blood glucose changes of mice in the different treatment groups.

Group	Blood Glucose (mmol/mL)	
	0 d	28 d
CK	5.95 ± 1.01	4.65 ± 0.98
NC	19.91 ± 1.27 **	23.92 ± 3.86 **
SP	19.97 ± 1.57 **	15.98 ± 2.03 **##
HEL	19.88 ± 1.10 **	19.64 ± 1.17 **##
HEM	19.92 ± 2.68 **	17.89 ± 2.22 **##
HEH	18.97 ± 1.31 **	17.16 ± 1.56 **##

CK: Blank control group; NC: Model control group; PC: Sitagliptin group; HEL: Active peptide low-dose group; HEM: Active peptide medium-dose group; HEH: Active peptide high-dose group; compared with the CK group, ** *p* < 0.01; compared with the NC group, ## *p* < 0.01.

**Table 7 foods-11-02148-t007:** Glucose tolerance changes of mice in the different treatment groups.

Group	Sugar Tolerance (mmol·L^−1^)				AUC
	0 h	0.5 h	1.5 h	2 h	
CK	4.65 ± 0.98	7.20 ± 1.31	5.91 ± 1.93	5.20 ± 1.38	12.26 ± 1.90
NC	23.92 ± 3.86 **	30.24 ± 3.78 **	28.19 ± 5.74 **##	25.64 ± 3.71 **	55.45 ± 8.72 **##
SP	15.98 ± 2.03 **#	24.00 ± 3.47 **##	14.55 ± 4.77 **##	13.45 ± 7.19 **##	38.08 ± 7.95 **##
HEL	19.64 ± 1.17 **#	26.23 ± 2.88 **##	18.43 ± 1.63 **##	17.18 ± 2.71 **##	45.27 ± 6.70 **##
HEM	17.89 ± 2.22 **#	24.64 ± 2.98 **##	16.66 ± 4.24 **##	15.10 ± 4.70 **##	40.44 ± 8.73 **##
HEH	17.16 ± 1.56 **#	26.35 ± 3.92 **	16.07 ± 3.56 **##	12.97 ± 2.59 **##	40.37 ± 7.92 **##

CK: Blank control group; NC: Model control group; PC: Sitagliptin group; HEL: Active peptide low-dose group; HEM: Active peptide medium-dose group; HEH: Active peptide high-dose group; compared with the CK group, ** *p* < 0.01; compared with the NC group, # *p* < 0.05, ## *p* < 0.01.

## Data Availability

Data are contained within the article.

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
