# Peer review of "Identification of the DPP-IV Inhibitory Peptides from Donkey Blood and Regulatory Effect on the Gut Microbiota of Type 2 Diabetic Mice"

_foods, 2022, doi:10.3390/foods11142148_

Round 1

Reviewer 1 Report

I thank the authors for the effort made in the work entitled "Identification of the DPP-IV inhibitory peptides from donkey blood and regulatory effect on the gut microbiota of type 2 diabetic mice". I find the approach to the work very interesting, as well as the results reported. However, it is not clear why they chosed donkey blood as the matrix for the production of these peptides. This is not a sustainable approach when looking for an alternative to synthetic drugs. I recommend that they clarify this issue with emphasis. On the other hand, here are some other comments on issues that I have encountered throughout the manuscript:

Line 144: mentions a flow rate of 600 L/min, isn't it 600 microL/min?

Lines 145-146: Isn't there an error in the description of phase B in the following sentence "The mobile phase consisted of acetonitrile/formic acid/water (2/0.1/98, v/v/v/v) for buffer A and acid/water (80/0.1/20, v/v/v/v) for buffer B."?

Line 147: Missing punctuation after "(Table 1)". It is recommended that punctuation throughout the text be revised (e.g., line 315 is also missing the final functus of the paragraph). 

Table 6: In 0d, is there really a difference between SP and NC as marked with "#"?

Lines 321-322: There is an error in the following sentence "The improvement in pancreatic tissue structure was more noticeable in the SP, HEL, HEM, and HEH groups than in the HEM group." repeat the "HEM".

Line 350: There is an error in "...Rank-Abundance curve (Figure 6-C) curve..."

Line 389: Repeats the word "analysis" in "In the principal coordinates analysis (PCoA) analysis table (Figure 8B)".

Lines 431-444: In these lines of the Discussion section, the authors make an introduction rather than a discussion. They also present some errors in punctuation and in the use of connectors.

Lines 448-450: They use the same sentence 2 times.

Line 528: In the title where it should say Conclusion they put Discussion again.

Other issues: Much use of passive voice. It is recommended to reformulate the sentences with active voice. Figure 9 does not match with its citations in the text. The references in the figures are unreadable due to the small size and poor graphic quality. In Conclusions they do not conclude, but rather summarize the description of results as if it were an abstract. 

Author Response

Thank you very much for your recognition of the work of this paper. Thank you for your valuable advice. That means you read it very carefully. I really appreciate your contribution to this paper. Please see the attachment for detailed reply.

Reviewer 2 Report

The authors aim to identify novel therapeutic peptides for T2DM, extracted from donkey blood, and study their function in a mouse model of diabetes. The goal is to find new treatments for T2DM, derived from natural products, that cause less undesirable side effects. Specifically, they are trying to identify new DPP-IV inhibitors, which have been shown to direct their action against GLP1 and GIP.

This is a relevant story and the peptides identified could have potential applications in the field.

This is a topic that has seen growing interest, so in that sense it is not novel, yet always important to explore and identify new therapeutic modalities.

This study is exploring the function of the purified peptides by sequencing the microbiome of control and T2DM mice -STZ/HFD model- treated with them. The authors suggest the use of processed donkey blood/hemoglobin as functional food in the future.

The paper is generally well-written. It would only require some minor editing for English.

The paper is written in a very elaborate and easy to follow manner.

The experiments are well planned and executed. The authors addressed their main questions and presented their results in a comprehensive manner.  

Author Response

Thank you very much for your affirmation of our experiment and your review of our article. For the language part, we have found a professional language modification company to modify it.Thank you. Wish you all the best in your work and a happy life.